# Thermal Comfort Improvement with Passive Design Strategies in Child Development Centers in Thailand

**Apiparn Borisuit [1,2] and Phanchalath Suriyothin [1,2,*]**

1   Department of Architecture, Chulalongkorn University, Bangkok 10330, Thailand
2   Design for Creative Economy Development Research Unit, Chulalongkorn University, Bangkok 10330, Thailand
*   Correspondence: phanchalath.s@chula.ac.th; Tel.: +66-2-218-4302

**Abstract:** Child Development Centers (CDCs) in Thailand are developed from the same national standard building plan across the country. Due to hot weather conditions, low-cost building materials, and a failure to consider the specific surrounding conditions of each case, thermal discomfort results. This study focuses on an improvement in the thermal comfort of a pilot CDC building in Maha Sarakham province, Thailand. Three CIBSE TM52 model criteria were applied to assess the level of overheating in the CDC building. The IESVE simulation tool was employed to assess the improvement from using passive design strategies (such as orientation, solar protection, thermal insulation, and ventilation). The results showed that passive design strategies could improve the overall thermal comfort of the CDC building. Thermal insulation, especially roof insulation, was the key element in reducing overheating in the building. A fully insulated building with shading devices and a night-time only window-opening pattern could meet the three targeted overheating criteria. Although the limitations of using the CIBSE TM52 model in hot and humid regions have been identified, these findings can be used as an exemplar of passive design strategy integration for other CDC buildings across the country.

**Keywords:** thermal comfort; Thailand; passive design strategies; child development center

## 1. Introduction

Quality experiences in early childhood have a positive impact on the development of children. High-quality care and education are crucial during cognitive, learning, and social skills development—particularly in the first five years of early childhood [1–3]. To provide a level playing field for all early learners, the Thai government (under the Department of Local Administration, Ministry of Interior) has established a Child Development Center Program and Child Development Centers (CDCs) as a community facility. CDCs are small preschools that offer childcare at no cost to families where the parents are unavailable due to work obligations. The program and CDCs aim to fulfill the necessary educational preparation requirements for early childhood (3–5 years) development prior to primary school [4].

Most CDC buildings have been constructed from a national standard building plan supplied by the Department of Local Administration. The same standard building plan accounts for over 18,878 CDC buildings in the country [5], mostly in rural areas of Thailand. Using the same CDC standard building plan for the whole country, regardless of the thermal environment, has resulted in discomfort, leading to dissatisfaction with regard to thermal comfort [6]. Considering that each location has its own specific conditions, this requires a different building approach according to those specificities, such as unique surrounding environments, different regional climates, and other parameters relevant to building design. In addition, due to limited budgets and construction techniques, low-cost construction materials are specified in the standard plan, such as single-pane clear glass

windows and a roof system without insulation. These inefficient and inappropriate design elements can reasonably cause overheating and thermal discomfort issues.

Consequently, air-conditioning systems are often installed to improve occupant comfort, resulting in maintenance costs and monthly electricity bills. Thailand's national budget allocated to each local administrative organization pays for these avoidable additional expenses. As a result, the CDC national standard building plan is not sustainable in the long term. In brief, the design fails to include an awareness of energy efficiency or a reduction in $CO_2$ emissions, and this takes place on a national scale.

To conserve energy and lower $CO_2$ emissions, buildings should be constructed to adapt to climatic conditions and to use physics principles to enhance building performance. Passive design plays an important role in low-cost construction to create comfortable conditions inside buildings by making use of local climate and site conditions. According to the Köppen-Geiger climate classification system, Thailand's climate is mainly classified as a tropical wet-dry or savanna climate (Aw), with some areas identified as having a tropical monsoon climate (Am) [7,8]. Official records show there are three seasons in Thailand: summer is from mid-February to mid-May; winter is from mid-October to mid-February; and the rainy season, influenced by the southwest monsoon, occurs from mid-May to mid-October. During summer, the average high temperature ranges from 33 °C to 36.2 °C, with the highest temperature reaching 44.6 °C in northern Thailand, according to ten-year records from the Thai Meteorological Department [9]. Winter is influenced by the northeast monsoon, which brings cold and dry air from anticyclones in China. The average minimum temperature is from 17.5 °C to 23.2 °C; while in the rainy season, the average temperature range is 23.8 °C to 33.4 °C. Throughout the year, relative humidity remains high, especially in the rainy season (78–84%). As a result, these climatic conditions exceed the thermal comfort range, especially during summer [10,11].

However, concerning adaptation to thermal comfort, people in tropical countries tend to withstand higher temperatures more easily. Two field studies in Thailand have shown that under natural ventilation conditions, people can be comfortable at the upper limit of thermal comfort with temperatures reaching 31.5 °C [12,13]. Comparing the thermal comfort model for the same period, the ASHRAE standard indicated the summer thermal comfort zone as being between 22.8 °C and 26.1 °C [14]. Additionally, previous studies found a correlation between thermal comfort sensation and wind sensation [15]. One study [10] conducted a field study with 288 Thai volunteers to investigate the impact of air velocity on thermal comfort to explore the potential of using electric fans instead of air conditioners to achieve thermal comfort. Under non-air-conditioning circumstances, the higher the air speed, the higher the neutral temperatures voted for. This showed that the thermal comfort range could be further extended, especially when the outdoor temperature is higher than 30 °C, thanks to the additional air movement. The authors then developed a ventilation comfort chart for non-air-conditioned buildings within the limit of an air velocity of 3 m/s, and a maximum air temperature of 36.3 °C.

Accordingly, a fixed range of overheating limits in terms of temperature may not be the most suitable way to assess thermal comfort in a free-running building. Traditional steady-state thermal comfort models do not consider human adaptation to the environment. Furthermore, expectations, thermal history, and cultural background can influence the limits of thermal comfort [16,17]. The "adaptive" comfort model approach may be a more suitable alternative to evaluating thermal comfort in a naturally ventilated building, as it takes into account the prevailing mean outdoor temperature. The comfort boundaries can be wider with an increasing outdoor temperature. This adaptive approach to assessing thermal comfort in naturally ventilated buildings has become the basis for global thermal comfort standards and is commonly used to assess thermal comfort in a free-running mode, as seen in ANSI/ASHRAE 55 [18], EN 16798 [19], and CIBSE TM52 [20].

In fact, achieving thermal comfort in a free-running building in a tropical climate is quite challenging. In the case of Thailand, most parts of the country experience hot and humid weather throughout the year, and there is high potential for increasing the risk of

overheating in buildings. Heat gain in buildings is mainly caused by radiant heat gain from sun exposure, ventilation heat gain, conduction heat gain through building surfaces, and internal heat gain [21]. Due to heat gains and heat transfer, the indoor air temperature is usually higher than the outdoor temperature in Thailand [11]. Given that it is important to prevent heat gain from direct radiation and to flush out hot air from buildings, passive design strategies, such as sun protection, thermal insulation, and cross-ventilation, can help reduce overheating in buildings [11,22–24]. These strategies have the potential to be an effective solution to compromised thermal comfort in low-cost buildings [15,22,25–29], such as CDC buildings.

Therefore, this study focuses on using passive design strategies to improve thermal comfort in a CDC building in Thailand. A building in Maha Sarakham was selected for a pilot study at an existing location. The adaptive thermal comfort model was adopted for overheating assessment using computer simulation. The first objective was to assess the thermal comfort of the standard building plan. Second, to improve thermal comfort in the building, this study proposes passive design strategies, including different orientations, solar protections, thermal insulations, and ventilation. Each passive design strategy was evaluated and analyzed to exploit the potential for thermal comfort improvement. This research aims to propose recommendations for integrating passive design strategies into the uniform national standard building plan for CDCs to solve thermal comfort issues.

## 2. Methods

An existing CDC building was selected as a pilot study location to assess thermal comfort. A baseline model was established according to the standard building plan provided by the Department of Local Administration. Next, passive design strategies were integrated to improve thermal comfort. CIBSE TM52 (2013) overheating criteria [20] were used to assess thermal comfort. This section includes descriptions of the study area and its climatic conditions, building geometry, building materials, simulation tools, and passive design strategies, as well as the criteria adopted in the study.

### 2.1. Study Area and Weather Data

The northeastern region of Thailand was chosen for the study because it is the largest and most populous region in the country; almost one-third of the population in Thailand lives in the region [30]. Maha Sarakham province is situated in the center of northeastern Thailand. A CDC in "Ku Santarat" (name of the district) in Maha Sarakham was selected as the pilot study location for this research. The CDC is located at 16.11° N 103.18° E at an altitude of 152 m above sea level. The location is shown in Figure 1.

Overall, Thailand's climate is influenced by two monsoons: the southwest and northeast monsoons. The three distinct seasons are winter (from mid-October to mid-February), summer (from mid-February to mid-May), and the rainy season (from mid-May to mid-October). The northeastern region of Thailand falls within a tropical wet–dry climate (Aw), which leads to hot and humid conditions. The highest, lowest, and average temperatures of 2021 in Maha Sarakham were 40.3 °C, 11.6 °C, and 28 °C, respectively [31]. The monthly highest, lowest, and average temperatures, relative humidity, and wind characteristics are also presented in Table 1.

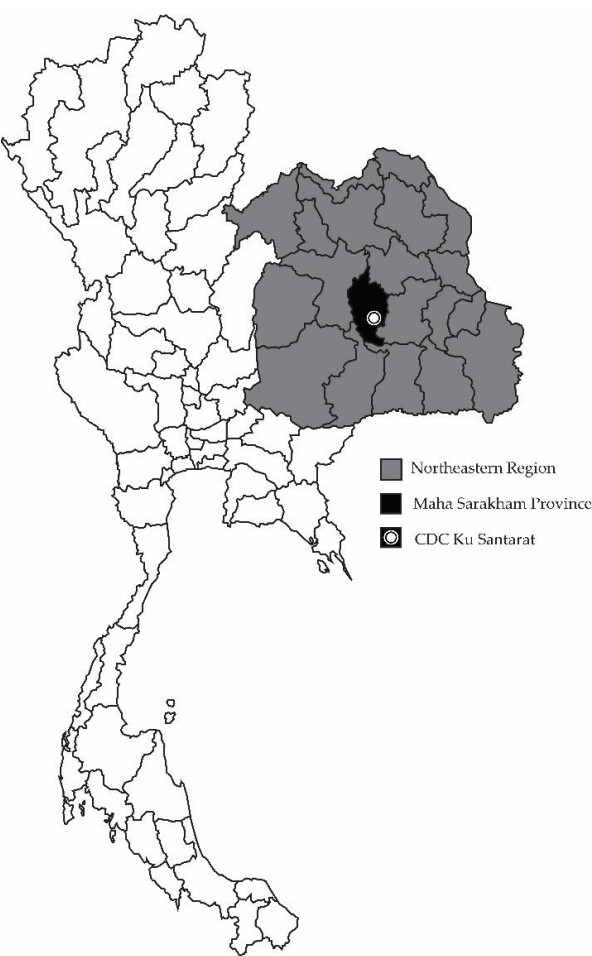

**Figure 1.** Map of Thailand showing the Maha Sarakham province and Child Development Center (CDC) locations.

**Table 1.** 2021 monthly climate data in Maha Sarakham [31].

| Month | Highest Temperature (°C) | Lowest Temperature (°C) | Average Temperature (°C) | Average Relative Humidity (%) | Average Wind Speed at 11.90 m (knot)/ Direction |
|---|---|---|---|---|---|
| January | 34.6 | 11.0 | 22.41 | 66.85 | 2.9/NE |
| February | 38.0 | 15.8 | 25.78 | 67.45 | 2.07/NE |
| March | 39.5 | 19.0 | 29.51 | 72.15 | 1.30/SE |
| April | 39.0 | 23.5 | 30.31 | 68.05 | 1.70/SE |
| May | 36.9 ** | 18.0 ** | 28.50 ** | 76.1 ** | 2.20/SW ** |
| June | 40.0 | 23.0 | 30.60 | 72.48 | 2.11/SW |
| July | 38.3 | 23.0 | 29.72 | 76.11 | 2.54/SW |
| August | 37.7 | 23.0 | 30.02 | 74.52 | 1.78/SE |
| September | 35.0 | 23.0 | 28.20 | 82.42 | 1.78//SE |
| October | 35.3 | 20.6 | 27.89 | 79.94 | 1.61/NE |
| November | 34.8 | 17.6 | 26.70 | 70.55 | 2.04/NE |
| December | 34.4 * | 14.5 * | 24.10 * | 69.34 * | 2.09/NE * |

** 2022 climate data; * 2020 climate data (due to the missing 2021 data).

### 2.2. CDC Building and Simulation Tools

The Department of Local Administration distributed three types of standard building plans for the construction of CDCs: (1) small size for less than 50 children; (2) medium size for 50–80 children; and (3) large size for more than 100 children. The selected pilot study, medium size, is a single-story building, consisting of one multi-purpose room and two separated rooms: an infirmary and a storage room. There is also a small building providing a kitchen and a toilet area behind the main building. In this study, only the main building area was targeted for the thermal comfort assessment.

CDC opening hours are from 8 a.m. to 3 p.m. When children arrive at the CDC, they spend their morning in the multi-purpose room to learn or participate in activities with teachers. They occasionally play outdoors in good weather. They then have lunch in the kitchen at noon. Early afternoon is naptime; children make their beds and sleep in the multi-purpose room. After that, they wake up and prepare to go back home around 3 p.m.

Figure 2 shows the standard building plan and elevations of a medium-sized CDC building. According to the standard building plan's specifications, a reinforced concrete structure and infill brickwork walls will be used. The sliding windows are set at dimensions of 1.50 m × 1.50 m and 2.50 m × 1.50 m, located one meter above the floor, which is the standard window size in Thailand. Table 2 describes the characteristics of the materials according to the standard plan specifications.

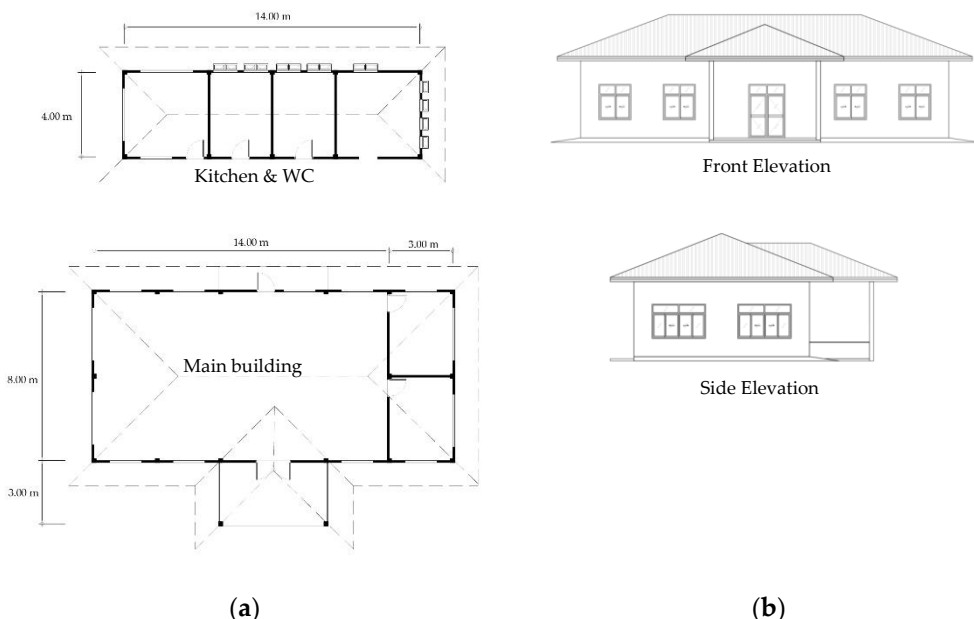

(**a**)  (**b**)

**Figure 2.** Drawings of the medium-sized CDC standard plan: (**a**) plan; (**b**) elevation.

**Table 2.** Materials specified in the standard drawing.

| Architectural Element | Materials | Thickness (mm) | U-Value (W/m²K) |
|---|---|---|---|
| Roof | Asbestos cement tile | 5.5 | 6.55 |
| Ceiling | Gypsum board | 9 | 2.94 |
| External wall | Brickworks | 100 | 3.89 |
| External windows | Clear float glass | 6 | 6.31 |
| Partition | Light wall system | 125 | 3.606 |
| Floor | Concrete with ceramic tile | 125 | 3.86 |

A computer simulation of the CDC building was conducted using dynamic thermal simulation in IESVE (Integrated Environmental Solution—Virtual Environment) software version 2021.1.1.0 [32]. A simple model of the standard CDC and the kitchen building was built from the drawing in Figure 2 and the materials in Table 2. To simulate the virtual environment, a typical meteorological year (TMY) generated from the Photovoltaic Geographical Information System (PVGIS) tool [33], accounting for the years between 2007 and 2016, was used. A total of 10 people were added to the simulation. The building was modeled as being naturally ventilated without mechanical cooling. The air exchange rate (ACH) was set at 1.2 h$^{-1}$ according to the calculation using Equation (1), in which $Q_v$, or volume flow rate, was calculated from Equation (2):

$$ACPH\left(h^{-1}\right) = \frac{Q_V\left(m^3/h\right)}{V\left(m^3\right)} \tag{1}$$

$$Q_v = C_d A_w u_r \left(\Delta C_p\right)^{0.5} \tag{2}$$

where $C_d$ represents the discharge coefficient for an opening; $A_w$ is the equivalent area of openings; $u_r$ stands for reference wind speed; $C_p$ denotes surface pressure coefficient; and $V$ is room volume.

To integrate natural ventilation for running the simulation, in the first part of the study, the windows were opened only during the hours of occupation. During the off-hours, when no one used the building, the windows were closed. Hereafter, in the ventilation analysis (Section 3.3.3), the study explores thermal performance using different window-opening patterns, such as daytime and night-time opening, opening only in the morning, and opening only during night-time (night cooling). Using the Apache and Vista Pro tools in IESVE, an overheating assessment was calculated. The simulation was run from 8 a.m. to 3 p.m. in accordance with the CDC's operating hours.

### 2.3. Passive Design Strategies

Selected passive design strategies were integrated to improve thermal comfort in consideration of reducing heat gain from solar radiation and promoting natural ventilation, which are the principles of tropical climate design. Strategies such as building orientation, shading devices, insulation, and additional ventilation were selected. First, the building orientation model was rotated by 45 degrees around the Z-axis from north to northwest to study the impact of orientation. Second, each abovementioned passive design strategy was applied: (1) the solar protection strategy includes overhangs and external shades (shutters), which are lowered when incident radiation is more than 300 W/m$^2$ or during hours of occupation; (2) thermal insulation is used in the roof, ceiling, internal walls, external walls, and insulated glass; and (3) the ventilation strategy comprises roof ventilation underneath the eaves and the use of night ventilation. All the passive design strategies used in this study are summarized with a U-value of the elements in Table 3. Figure 3 illustrates all the passive design strategies employed.

**Table 3.** Passive design strategies to improve thermal comfort in the CDC.

| Strategies | Description | |
|---|---|---|
| (A)  Orientation | 8 orientations: existing condition (315° N), 0° N, 45° N, 90° N, 135° N, 180° N, 225° N, 270° N | |
| (B)  Solar protection | | |
| (B1) Overhang | 1.2 m overhang projection + 0.5 m left and right projection | |
| (B2) Shutter 300 | Lower shutters when incident radiation > 300 W/m$^2$ | |
| (B3) Shutter occupied | Lower shutters during occupied hours | |
| (C)  Thermal insulation [1] | | U-value (W/m$^2$K) |
| (C1) Roof insulation | 150 mm thick polyurethane board | 0.1625 |
| (C2) Ceiling insulation | 150 mm thick polyurethane board + 12 mm gypsum board | 0.1588 |
| (C3) Wall insulation | 15 mm cement plaster + 75 mm lightweight concrete block + cavity + 65 mm thick polyurethane board + 12 mm gypsum board | 0.3055 |
| (C4) Partition | 12 mm gypsum board + cavity + 65 mm glass fiber slab + 12 mm gypsum board | 0.381 |
| (C5) Insulated glass | 6 mm clear glass + cavity + 6 mm clear glass | 2.03 |
| (D)  Ventilation | | |
| (D1) Roof ventilation | 50% openable area grille under eaves soffit | |
| (D2) Additional windows for night cooling | -    Opening windows at night<br>-    Additional top-hung windows above existing windows | |

[1] Outside to inside.

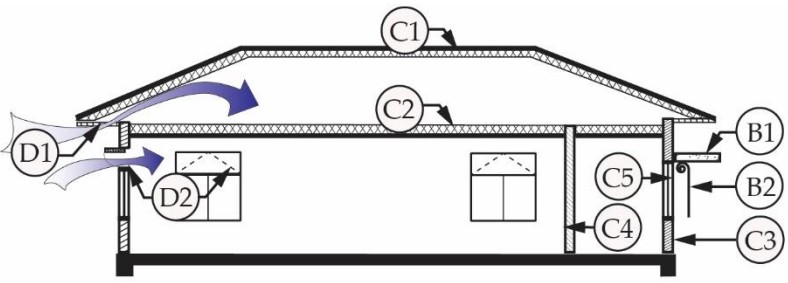

**Figure 3.** Illustration of all passive design strategies.

The level of overheating was assessed to identify the improvement in thermal comfort for each strategy. After incorporating each strategy investigation into the design, different patterns of window opening were applied to the simulation, which included daytime and night-time, occupied hours, morning only, and night-time only.

### 2.4. Overheating Assessment Criteria

An adaptive approach to thermal comfort reveals that an acceptable indoor temperature in free-running buildings is related to outdoor temperatures. In this study, thermal comfort was assessed according to the CIBSE TM52 overheating criteria [33]. This study aims to determine an existing occupied building, following the methodology and recommendations of BS EN 15251 (BSI, 2007) [34].

As comfortable temperature varies with outdoor temperature, the maximum acceptable indoor temperature ($T_{max}$) can be calculated from ($T_{rm}$), as shown in Equation (3):

$$T_{max} = 0.33\ T_{rm} + 21.8 \tag{3}$$

where $T_{max}$ is the maximum acceptable temperature (°C); and $T_{rm}$ represents the running mean of the outdoor temperature (°C).

The most important factor to determine the risk of overheating is $\Delta T$, which is the difference between the actual operative temperature ($T_{op}$) in the room at any time ($T_{op}$) and $T_{max}$, as calculated in Equation (4):

$$\Delta T = T_{op} - T_{max} \tag{4}$$

Table 4 summarizes the three criteria used to assess the risk of overheating. A room or building that fails any two of the three criteria is classified as overheating.

**Table 4.** Overheating assessment criteria according to CIBSE TM 52.

| | Assessment Criteria | Acceptable Deviation |
|---|---|---|
| Criterion 1: | Percentage of occupied hours during which $\Delta T \geq 1$ K | <3% of occupied hours |
| Criterion 2: | Daily weighted exceedance ($W_e$) > 6 degree h/day | 0 day |
| Criterion 3: | Maximum temperature level ($T_{upp}$): $\Delta T \geq 4$ K | 0 h |

As stated in criterion 1, the number of occupied hours that the operative temperature exceeds the threshold comfort temperature by 1 K must be less than 3%. Criterion 2 indicates the severity of overheating. Daily weighted exceedance ($W_e$) should be less than or equal to 6 on any day. $W_e$ can be calculated as shown in Equation (5):

$$W_e = \left(\sum h_e\right) \times WF = (h_{e0} \times 0) + (h_{e1} \times 1) + (h_{e2} \times 2) + (h_{e3} \times 3) \tag{5}$$

where $WF$ is the weighting factor. $WF$ must be 0 if $\Delta T$ is less than or equal to 0; otherwise, $WF$ is equal to $\Delta T$. $h_{ey}$ is the time ($h$) when $WF$ is $y$.

For criterion 3, the absolute maximum value for the indoor operative temperature must not exceed 4 K. This threshold is the upper limit temperature for restoring personal comfort.

## 3. Results

### 3.1. Baseline Model

Initially, a CIBSE TM52 overheating assessment was carried out for the baseline model. The results show that the building constructed using the standard building plan experiences extreme overheating. The baseline model fails all three criteria: (1) the number of hours of exceedance is over 33.9%, and the baseline model's hours of exceedance are 11 times higher than the limit; (2) the $W_e$ of the baseline model is equal to 25, so the building does not pass the criterion; and (3) the maximum value for the indoor operative temperature is 7 K, which is much more than the 4 K limit. The baseline model does not achieve criteria I, II, or III.

Figure 4 shows the calculated values according to criteria I, II, and III. The limits of each criterion are presented in dotted lines.

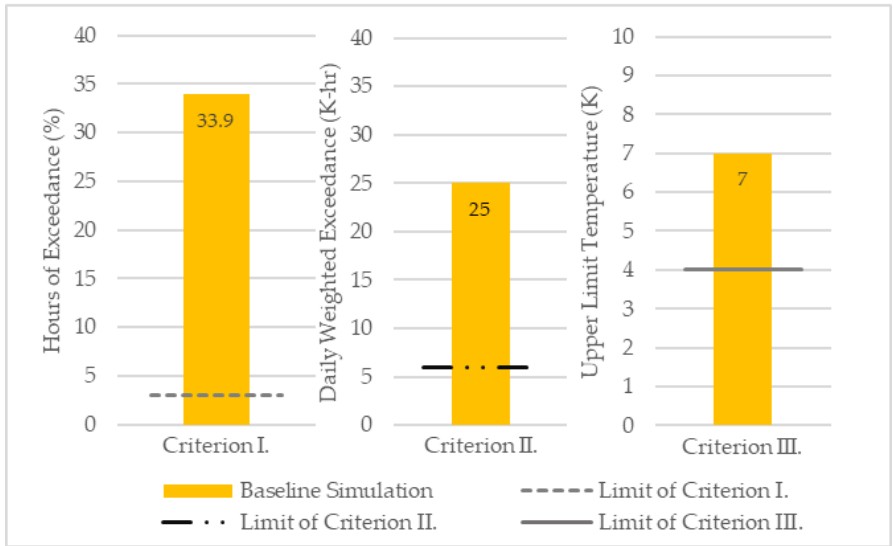

**Figure 4.** Results of overheating assessment criteria for the baseline model.

*3.2. Orientation*

The pilot building in Ku Santarat, Maha Sarakham, is northwest facing, while simulations of different orientations were used to compare thermal comforts, as shown in Table 5. Rotating the orientation of the building does not significantly improve thermal comfort. To assess the overheating criteria, the following data are presented: (1) the calculated hours of exceedance are 33–34.3%, which are not distinguished from the existing condition; (2) the daily weighted exceedance is slightly reduced from 25% to 23% with 225° N and 270° N; and (3) the upper limit temperature at 7 K still exceeds the limit.

**Table 5.** Results of overheating assessment with different orientations.

| Image | Degree from North | Criterion | | |
| --- | --- | --- | --- | --- |
| | | I. (%) | II. (K-hr) | III. (K) |
| | CDC Pilot study 315° N | 33.9 | 25 | 7 |
| | 0° N | 34.3 | 25 | 7 |
| | 45° N | 33.7 | 25 | 7 |
| | 90° N | 33.6 | 25 | 7 |

**Table 5.** *Cont.*

| Image | Degree from North | Criterion | | |
|---|---|---|---|---|
| | | I. (%) | II. (K-hr) | III. (K) |
| | 135° N | 34.2 | 25 | 7 |
| | 180° N | 34.1 | 25 | 7 |
| | 225° N | 33.6 | 23 | 7 |
| | 270° N | 33 | 23 | 7 |

*3.3. Passive Design Strategies*

Passive design strategies, as mentioned in Section 2.3, were integrated into the baseline model. Figure 5 illustrates the number of exceeded hours (criterion I), the daily weighted exceedance (criterion II), and the upper limit temperature (criterion III) simulated from each strategy separately and jointly as a combined strategy. It appears that using all the passive design strategies together could achieve criterion III. However, the other two criteria were not achieved.

Overall, it was evident that thermal insulation was the most effective strategy. Using insulation can reduce the hours of exceedance by 57.5%. Solar protection also helps to minimize hours of exceedance by 20%. Apparently, a building equipped with solar protection and insulation (strategy B + C) has a good performance that is equal to the performance when applying all strategies (strategy B + C + D). Both can reduce 70% of the hours of exceedance, 48% of the daily weighted exceedance, and 3 K of the maximum temperature. The ventilation strategy may not vastly improve thermal comfort in a room compared with the other strategies. Additionally, further investigation into ventilation was carried out. Other strategies were also thoroughly investigated in detail to understand the mechanisms of thermal comfort improvement.

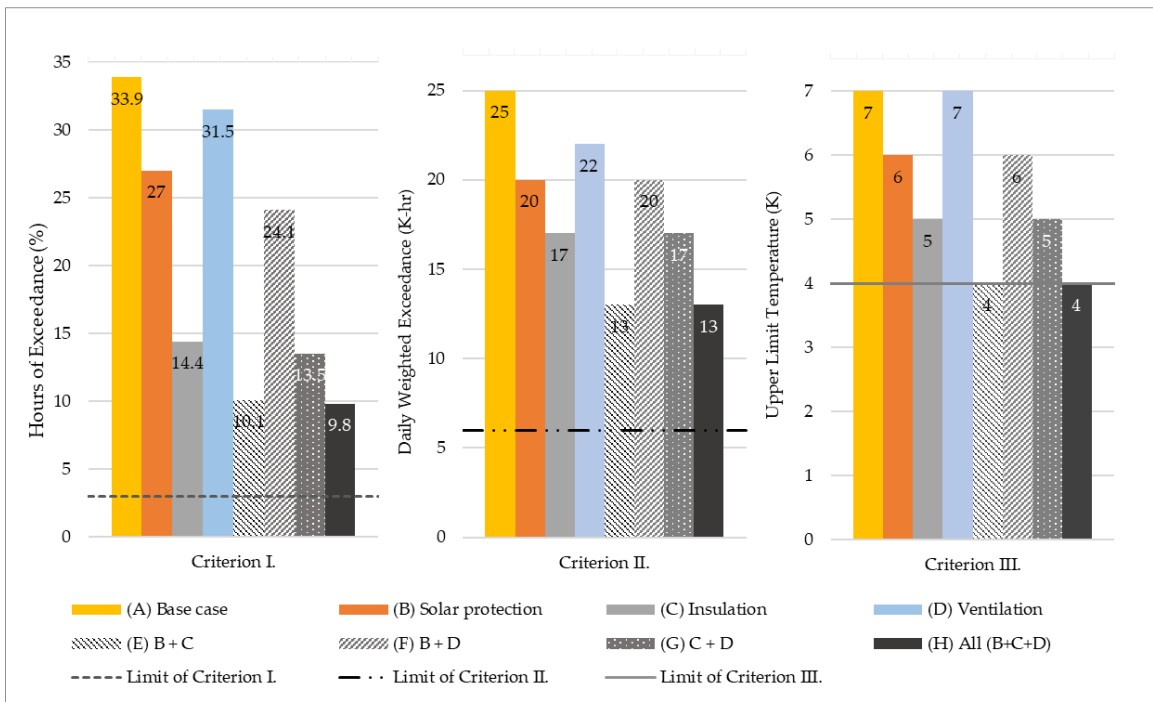

**Figure 5.** Results of overheating assessment using all passive design strategies.

### 3.3.1. Solar Protection

Each element, such as overhangs and shutters, was tested individually to assess the performance of the solar protection strategy. The simulation of shutters was carried out with two scenarios: (1) the shutters are lowered when incident solar irradiance is more than 300 W/m$^2$; and (2) the shutters are lowered during occupied hours. Figure 6 shows the overheating assessment criteria for this strategy. Installing overhangs can slightly improve thermal comfort more than lowering shutters when there is high solar gain, but the best way to improve thermal comfort is to lower the shutters during occupied hours. This can reduce thermal comfort factors, such as hours of exceedance and daily weighed exceedance by 17.8% and 20%, respectively. The maximum temperature was also reduced by 1 K.

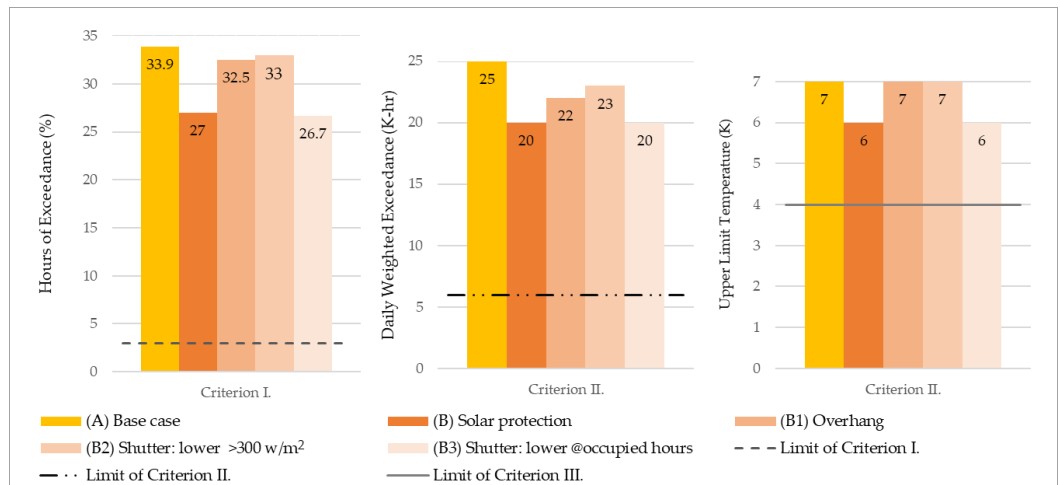

**Figure 6.** Results of overheating assessment using the solar protection strategy, including overhangs and shutters.

### 3.3.2. Thermal Insulation

Overall, the thermal insulation strategy can enhance thermal comfort more than the other strategies. Figure 7 demonstrates the calculated overheating criteria for each application of insulation, such as roof, ceiling, wall, and insulated glass. Both the room and roof spaces were assessed. The results from breaking down the simulation show that roof insulation is the most effective way to reduce the temperature of the room and under the roof. Moreover, ceiling insulation can also bring down the temperature of the room, but it heats up the roof space. Insulated glass can prevent solar gain from the windows, but heat gain from the roof cannot be avoided. As a result, the insulated walls keep the room warmer. Additionally, the insulated partition dividing the rooms did not improve thermal comfort in this case.

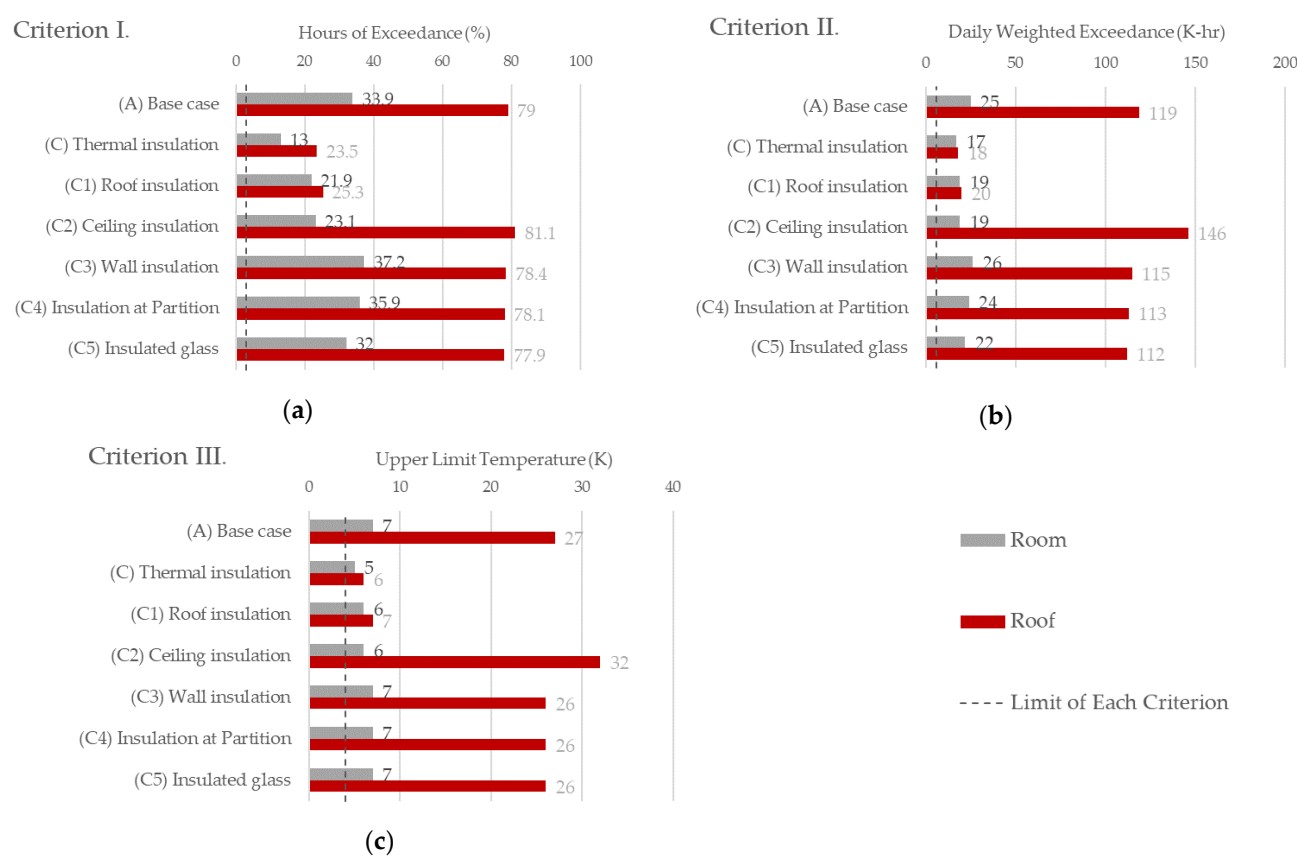

**Figure 7.** Results of overheating assessment of the room and roof area using the thermal insulation strategy: (**a**) criterion I; (**b**) criterion II; and (**c**) criterion III.

### 3.3.3. Ventilation

Figure 8 shows the results of the overheating assessment of the room and roof spaces using the ventilation strategy. These results indicate that roof ventilation helps to minimize some heat under the roof space when it lacks roof insulation. Despite additional openings to create more cross-ventilation, night windows do not prevent heat gains in the room.

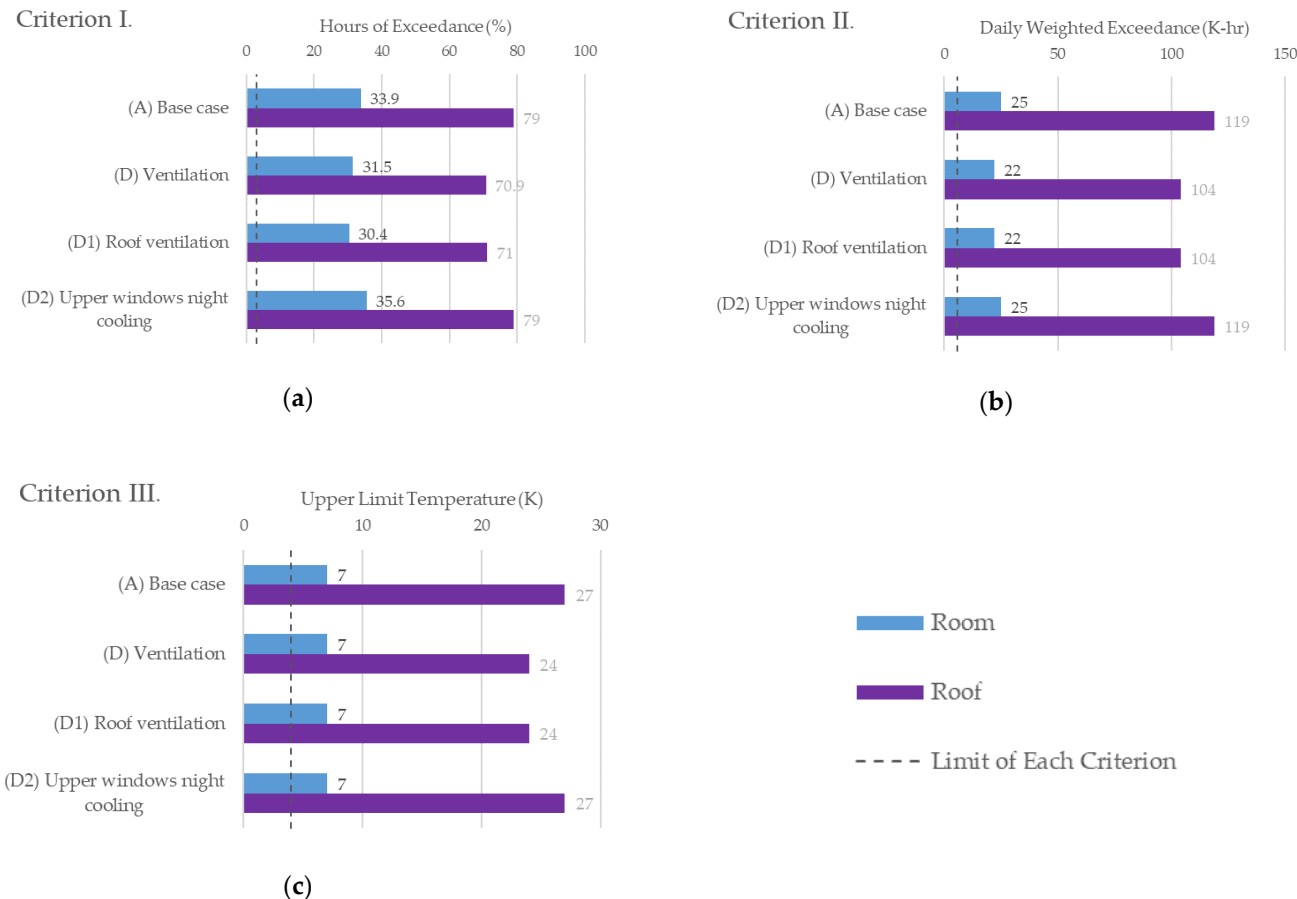

**Figure 8.** Results of overheating assessment of the room and roof area using the ventilation strategy: (**a**) criterion I; (**b**) criterion II; and (**c**) criterion III.

Cross-ventilation is a key passive design strategy in hot and humid conditions. Window-opening patterns are also important for appropriately adopting natural ventilation into the building. The study applied different patterns of window opening to the simulation with all passive design strategies. The patterns of window-opening include daytime and night-time, occupied hours, morning only, and night-time only. Table 6 shows the overheating assessment results with different window-opening patterns. The highlighted cells represent the strategies that meet the CIBSE TM52 criteria. The results reveal that the building with all strategies, opening the windows only during night-time could achieve two of the three criteria according to CIBSE TM52, while the building with solar protection and full insulation, opening the windows only during the night-time could achieve all criteria. Nonetheless, there is a concern about possible heat gain from the roof and enclosure. When heat is not effectively prevented by solar protection or insulation, extreme overheating can result, as shown in Table 6, under strategies A, B, C, D, F, and G at night-time only.

**Table 6.** Results of overheating assessment using all strategies with different window-opening patterns. The highlighted cells meet the targeted criteria.

| Opening Patterns | Daytime-Night-Time | | | Occupied Hours | | | Morning Only | | | Night-Time Only | | |
|---|---|---|---|---|---|---|---|---|---|---|---|---|
| Criterion: | I. | II. | III. | I. | II. | III. | I. | II. | III. | I. | II. | III. |
| (A) Base case | 33.2 | 23 | 7 | 33.9 | 25 | 7 | 39.8 | 31 | 12 | 58.6 | 46 | 13 |
| (B) Solar protection | 26.3 | 20 | 6 | 27 | 20 | 6 | 33.7 | 25 | 10 | 46.5 | 35 | 11 |
| (C) Insulation | 12.2 | 17 | 5 | 14.4 | 17 | 5 | 23.1 | 18 | 6 | 31.6 | 19 | 6 |
| (D) Ventilation | 31.1 | 22 | 7 | 31.5 | 22 | 7 | 38.3 | 31 | 12 | 57.4 | 45 | 13 |
| (E) B + C | 9.8 | 13 | 4 | 10.1 | 13 | 4 | 17 | 18 | 5 | 2.5 | 5 | 2 |
| (F) B + D | 24.1 | 19 | 6 | 24.1 | 20 | 6 | 30.8 | 23 | 9 | 41.3 | 31 | 10 |
| (G) C + D | 13.4 | 17 | 5 | 13.5 | 17 | 5 | 22.3 | 18 | 6 | 37.4 | 22 | 7 |
| (H) All strategies | 9.8 | 13 | 4 | 9.8 | 13 | 4 | 8.5 | 12 | 4 | 3.3 | 6 | 3 |

*3.4. Time Analysis*

To understand the causes and patterns of overheating in the building, a specific time was set for the overheating study. The TM 52 criteria employs two components to assess overheating: (1) degree exceeded adaptive temperature (in unit K); and (2) daily weighted exceeded temperature (in unit degree hours/day). First, the overheating criteria were investigated across the year. This helped us understand the seasonal impact of overheating. Second, we focused on the hottest day, 23 April, where the outdoor dry-bulb temperature peaked at 38 °C. Then, a 24-h interval of overheating was investigated to examine the impact of window-opening and shading device patterns.

3.4.1. Seasonal Effect

Figure 9 shows the degree exceeded adaptive temperature simulated using all passive strategies. The colored lines illustrate different pattern results: the blue line represents the temperature during the night-time window-opening pattern; the orange line denotes the temperature during the occupied hours window-opening pattern; the dotted line shows 1 K above the adaptive temperature, which is the limit of criterion I; and the dash-dotted line stands for the limit of criterion III at 4 K.

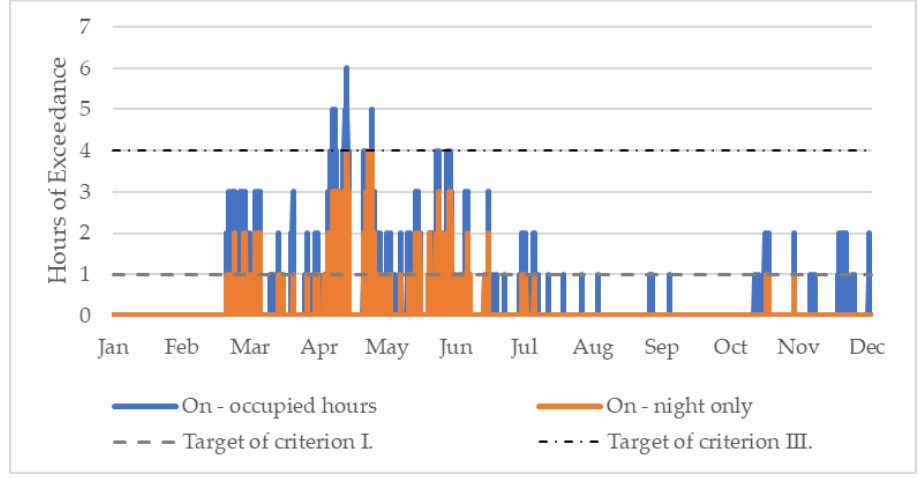

**Figure 9.** Degree exceeding adaptive temperature simulated using all passive strategies across the year.

Overall, it is evident that window-opening during occupied hours causes more overheating in the building. Although the difference between the operative temperature and the maximum acceptable temperature ($\Delta T$) is greater than 1 K under both conditions, $\Delta T$ is greater than 1 K during the summer months only (mid-February to mid-May) with night cooling. Otherwise, $\Delta T$ is greater than 1 K all year round with daytime window-opening patterns. Moreover, $\Delta T$ exceeds the upper limit (4 K) with daytime window-opening, while $\Delta T$ does not exceed 4 K with night cooling, despite the summer temperatures. Given that the calculation was conducted during the May–September period (a typical non-heating season in European countries, according to CIBSE TM52), the hours of exceedance from March to April are not included in the calculation. However, Thailand has different climatic conditions; its non-heating season is all year round. Criterion I must be recalculated based on the data for the whole year to assess the overheating of the buildings in Thailand.

Figure 10 illustrates the daily weighted exceedance. The colored lines illustrate different pattern results: the blue line represents the night-time window-opening pattern; the orange line denotes the daytime window-opening pattern; and the dotted line stands for 6 °C hr, which is the limit of criterion II. Again, window-opening during occupied hours causes more daily weighted exceedance.

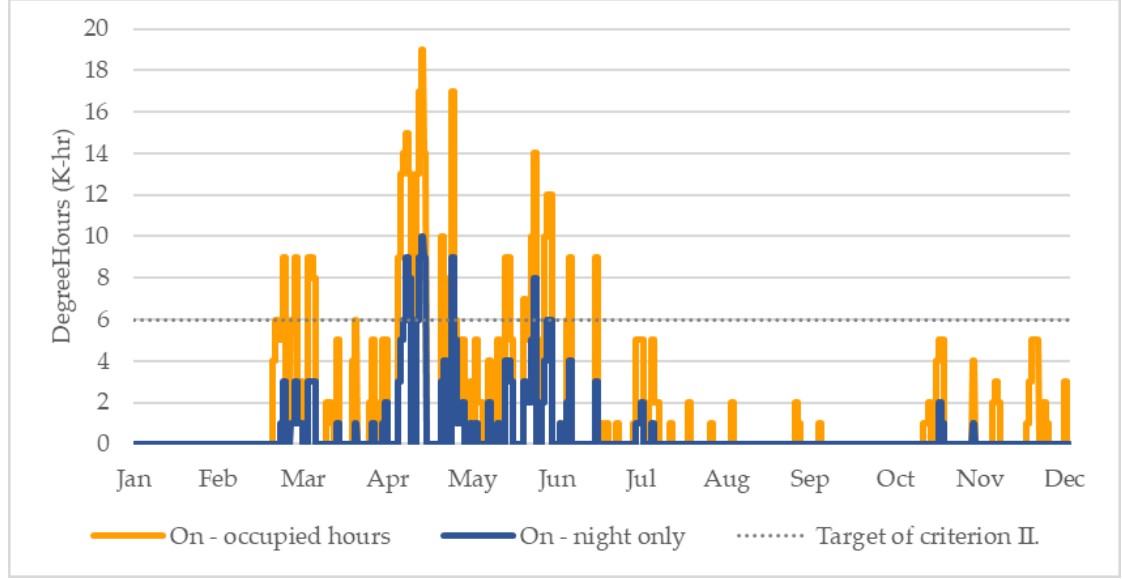

**Figure 10.** Daily weighted exceedance simulated using all passive strategies across the year.

From the simulation, applying all strategies with night cooling improved thermal comfort when compared with opening during occupied hours. The chart shows the exceedance during the summer months in Thailand, which is in accordance with Figure 9. It should be noted that the chart displays the data of 24-h, which includes the hours of non-operating hours. When the hours of exceedance of non-operating hours are included, the daily weighted exceedance ($W_e$) can be higher than the calculation shown in Table 6.

3.4.2. 24-h Interval

Four patterns of window-opening were observed for a 24-h interval on 23 April, the hottest day. The simulated models were equipped with all of the passive design strategies. The degree exceeding the adaptive temperature is shown in Figure 11. The blue, orange, yellow, and green circles represent the night-time, occupied hours, morning, and 24-h window-opening patterns, respectively. The dotted gray line denotes outdoor dry-bulb temperatures. The dotted yellow line and the dashed-dotted yellow line are direct and diffused radiation (in W/m$^2$).

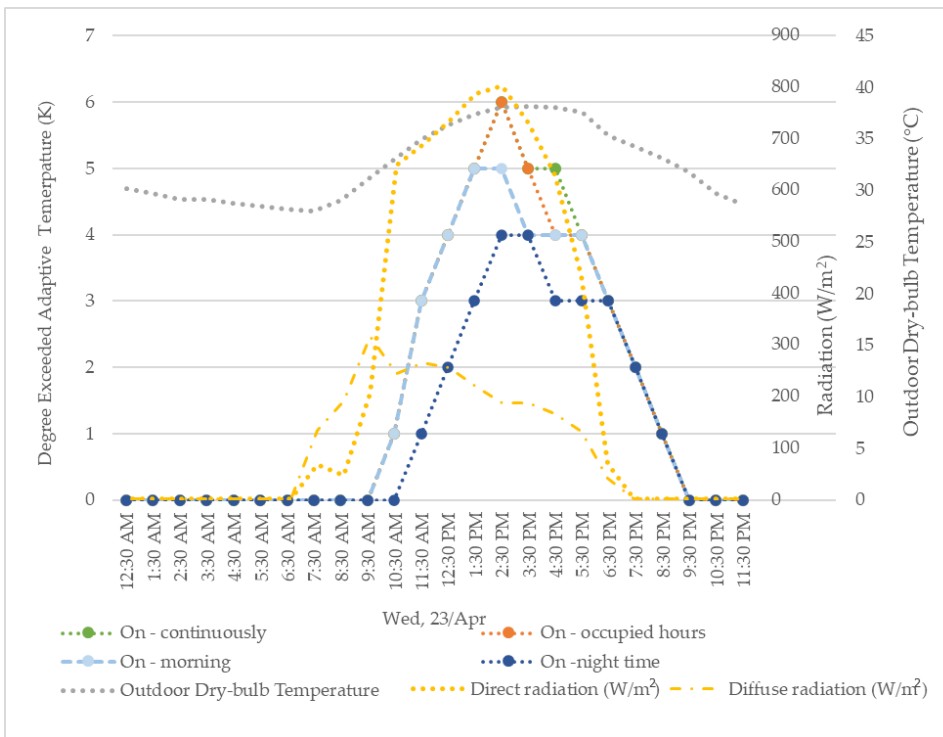

**Figure 11.** Degree exceeding adaptive temperature simulated using all passive strategies across the hottest day.

Temperatures rise across the morning, peak at 2:30 p.m., and decline in the evening. Clearly, $\Delta T$ throughout the day is in accordance with the direct radiation and outdoor temperature, as shown in the yellow and gray dotted lines. It is apparent that night-time opening can retain the heat from outdoors longer; $\Delta T$ starts rising after 10:30 a.m., while it rises after 9:30 when the windows are opened during the daytime. With the night-time opening pattern, the maximum temperature peaks at 4 K, while it peaks at 5 K and 6 K with daytime opening patterns. Closing the windows in the afternoon helps to reduce the temperature by at least 1 K for four hours, compared with opening during occupied hours or opening all day and night.

As solar gain has a great impact on overheating, this study further focused on solar protection strategies. Therefore, different shading devices were investigated on 23 April. Figure 12 shows the degree exceeded adaptive temperature of overhangs, shutter lowering when solar radiation is higher than 300 W/m², shutter lowering after 1 p.m., and shutter lowering during occupied hours. The dotted gray line represents the outdoor dry-bulb temperatures. The dotted yellow line and the dashed-dotted yellow line are direct and diffused radiation (in W/m²), respectively.

Temperatures rise across the morning, peak at 2:30 p.m., and decline in the evening, which is in accordance with Figure 11. Installing overhangs and lowering the shutters when solar radiation is higher than 300 W/m² can slightly reduce the degree exceeding the adaptive temperature in the early afternoon. Lowering the shutters in the morning can delay solar heat gain in the room during the morning. However, lowering the shutters during occupied hours can best minimize the temperature.

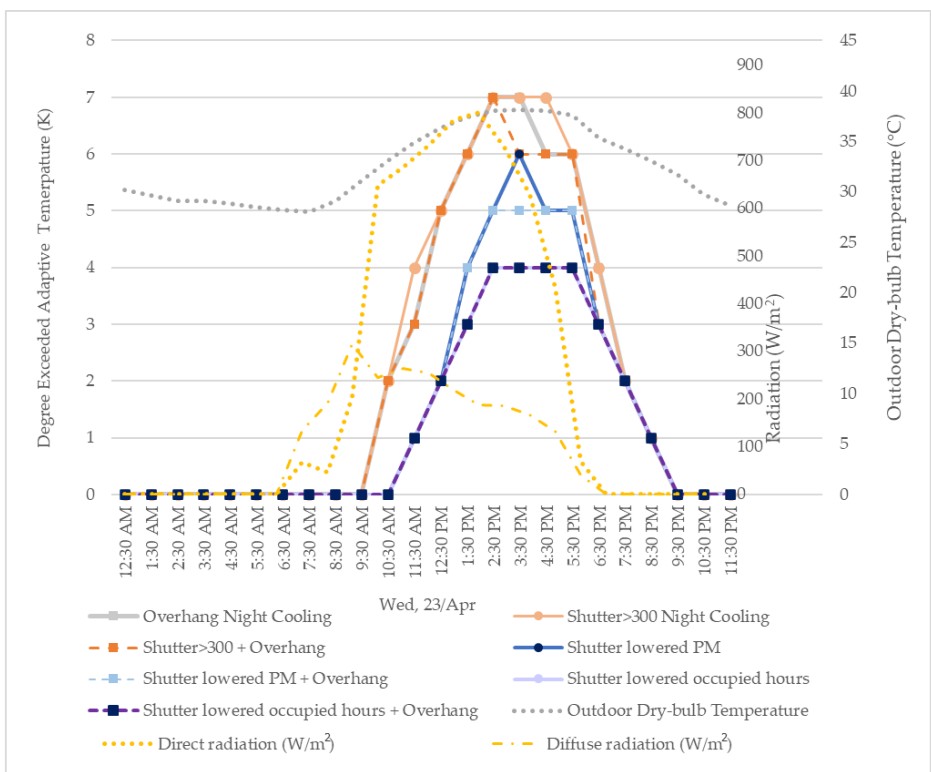

**Figure 12.** Degree exceeding adaptive temperature simulated using overhangs and shutters with different patterns across the hottest day.

## 4. Discussion

The results show that passive design strategies could substantially improve thermal comfort in CDC buildings. The key element to reducing overheating is thermal insulation, especially roof insulation. Remarkably, preventing hot air from entering from outside by opening the windows only at night-time greatly minimizes overheating, particularly when the building is equipped with full insulation and shading devices. These strategies help the building meet the targeted overheating criteria.

Thailand reports high average outdoor temperatures, high humidity, and moderate solar radiation. As temperature, relative humidity, and mean radiant temperature are the environmental factors of thermal comfort [18], the conditions in Thailand mean there is a high risk of overheating. It should be noted that the CIBSE TM52 standard was implemented for use in European countries in a high-latitude context. Its application in other countries where the temperatures are above 30 °C may not be practical [35]. It was developed from surveys with specific climatic data that differ from the tropical context in Thailand. Besides the high temperatures, experiencing high humidity also causes discomfort, but the operative temperature, as used in the CIBSE TM52 criteria, does not take into account the water content in the air. A standard that considers relative humidity a factor of thermal comfort will help in the assessment of the thermal sensation of occupants [36]. Furthermore, overheating criteria employ only "non-heating periods" to calculate the factors of risk of overheating. In European countries, the period from May to September was identified as non-heating, and only the exceedance hours during these five months were considered. On the other hand, the non-heating period in Thailand is likely to occur all year round. Periods of assessment must be reconsidered regarding the seasons in Thailand. Unlike schools, CDCs have no school breaks. Overheating in CDCs in Thailand must be assessed in all seasons. Taken together, although the current overheating assessment criteria were useful for evaluating overtime thermal comfort in this study, appropriate assessment criteria for hot humid regions are clearly needed in future studies.

Orientation is one strategy used to prevent prolonged radiation exposure. The best practice for planning a building in Southeast Asian countries is to be oriented east–west, which allows the longest sides of the buildings to face north and south. In this case, facing north–south or east–west is not distinguished in terms of thermal comfort. It is possible that both long-side façades have adjacent structures to protect them from sun exposure, such as the kitchen building and the entrance foyer. The best scenario is 270° N, where the kitchen located east of the main building protects it from sun exposure in the morning. The solar heat from the west may increase in the afternoon, but it peaks in the late afternoon when the occupants leave the building.

The results indicate that overheating in the building is related to the level of outdoor direct radiation. It is interesting that indoor overheating is more in accordance with direct radiation than with outdoor temperature. The principal areas that receive solar radiation are the roof surfaces due to the right angle and size of the surfaces compared with the other envelopes. This is why roof insulation is the most important factor. This conforms with previous studies on low-income housing in Thailand [22] and Uganda [25–27]. Note that the thickness of the roof insulation used in the simulation, as shown in Table 2, is not common in Thailand. The ideal insulation is intended to have a U-value close to the ASHRAE standard [37].

Roof ventilation cannot prevent heat transfer in the attic, but it can reduce this effect. The passive roof ventilation used in this study may not be the most optimum design for delivering a constant flow of cool air. Adding ridge venting or an active system, such as a roof turbine, is another solution for reducing more heat in the attic [38–40]. A double roof is another alternative to reducing overheating in the roof space [28,41]. However, this study aims to integrate passive design into the existing building design; a double roof was not integrated because it may have changed the architectural design.

Compared with roof surfaces, wall insulation is less effective. This is possibly due to less solar radiation on vertical surfaces. However, it should be known that transparent envelopes, such as windows, are the most sensitive material with a transmission property that permits solar radiation to enter, heat up, and be trapped. Insulated glass is one of the keys to retaining heat from outdoors [42]. It is also clear that covering window areas with shading devices can minimize the direct radiation going into the building. The results show that installing overhangs does not sufficiently reduce the solar gain that causes thermal discomfort. External shutters are a good alternative only when they are programmed to lower during occupied hours. This option is quite extreme because thermal comfort will be a trade-off with daylight availability, as it cuts off daylight into the room. Time analysis shows that the building starts to overheat over time and peaks around early afternoon, when it is nap time for children. It is reasonable to lower the shutters in the afternoon to protect solar heat gain. Since children are sleeping at that time, they do not require daylight in the building. Moreover, the additional top windows for night cooling might help provide daylight and ventilation across the day when the shutters are lowered. Overall, this study focused only on strategies that improve thermal comfort. A holistic approach to evaluating building performance must be integrated, using factors such as daylight quality and availability, ventilation, and energy consumption, as well as acoustic performance.

However, it should be considered that the building is occupied by sensitive and vulnerable groups, such as children. This group may fall under the standard suggested for Category I High level of expectation only used for spaces occupied by very sensitive and fragile persons [20,34], which sets the maximum acceptable temperature ($T_{max}$) at 2 K above the adaptive temperature [14]. In any case, some CDCs may be equipped with air-conditioning systems to enhance thermal comfort. A recent study of thermal comfort in elderly Thai people [43] suggested a guideline for enhancing thermal comfort and reducing energy consumption for multipurpose senior centers in Thailand. The use of natural ventilation, together with orbit fans in the morning and an air-conditioned mode in the afternoon, was recommended for the hot season in Thailand [43]. Integrating passive design, as recommended in this study, can help to minimize the energy used for

air conditioning due to its minimum use and the smaller difference between current and targeted temperatures. The optimum use of passive design elements must be further studied, especially in the analysis of return on investment.

**5. Conclusions**

This research aims to offer recommendations for using passive design strategies. It is evident that using thermal insulation, especially roof insulation, can minimize overheating. Night-time window-opening can prevent heat gain during the daytime and derive benefits from night cooling. It is also important to use shading devices; however, these must be used carefully, and daylight analysis must be carried out to ensure daylight performance when the shutters are lowered to protect them from sun exposure.

The limitations of using the CIBSE TM52 for assessment in Thailand have been specified. Future work regarding new regionally related criteria is required. Furthermore, it is important to consider future weather patterns in which temperatures may be higher according to climate change.

This research aims not only at improving thermal comfort for energy consumption and electricity bills, but also at increasing the quality of life of children and teachers in the CDC program. Children will have a better learning experience in a better environment. Combined, these can lead to an improvement in the three pillars of sustainability.

**Author Contributions:** Conceptualization, P.S. and A.B.; methodology, A.B.; software, A.B.; validation, A.B.; formal analysis, A.B.; investigation, A.B.; resources, A.B.; data curation, A.B.; writing—original draft preparation, A.B.; writing—review and editing, P.S. and A.B.; visualization, A.B.; supervision, P.S.; project administration, P.S.; funding acquisition, P.S. All authors have read and agreed to the published version of the manuscript.

**Funding:** This research project is supported by Ratchadapiseksompotch Fund Chulalongkorn University, 2021, in Society, Art, and Culture (2nd), for the project "Passive design strategies for occupants' comfort in children's development center using computer-based simulation" submitted by Phanchalath Suriyothin, head of the project.

**Institutional Review Board Statement:** Not applicable.

**Informed Consent Statement:** Not applicable.

**Data Availability Statement:** Not applicable.

**Acknowledgments:** The authors gratefully acknowledge Ratchadapiseksompotch Fund Chulalongkorn University for funding this research work through the project "Passive design strategies for occupants' comfort in children's development center using computer-based simulation" submitted by Phanchalath Suriyothin, head of the project. In addition, the authors are grateful for the support from Warakul Tantanatewin, Veerapong Eawpanich, Theeraporn Premchaiswadi, and Matchima Makka.

**Conflicts of Interest:** The authors declare no conflict of interest.

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
