# Peer review of "Thermal Comfort Improvement with Passive Design Strategies in Child Development Centers in Thailand"

_sustainability, doi:10.3390/su142416713_

Round 1

Reviewer 1 Report

In the proposed article, numerical simulations are used to evaluate and to compare several passive design strategies in order to reduce the thermal discomfort in a Child Development Center (CDC) in Thailand. It is underlined that most of the CDC in Thailand are constructed in the same way from a national standard building plan using low-cost materials which results in a recurrent overheating discomfort for the children. To avoid the use of air conditioning, the authors focus on passive strategies that perfectly fulfill the scope of the journal “Sustainability”. Three criteria from CIBSE TM52 model are used to quantify the discomfort for different scenarios. I appreciate that the authors clearly stated the limits of the chosen model that was initially proposed for European countries and not hot and humid weather like in Thailand. Although the chosen thermal comfort criteria do not integrate the level of relative humidity (that may be done in future studies), this work provides first practical and targeted recommendations for CDC:  insulation of the roof, solar protection and window opening at nighttime. Unfortunately, even considering this action plan we note that on hottest days, overheating in the building is attenuated but it is still over the considered criteria. After minor revisions, I recommend the article for publication in the journal “Sustainability”. I invite the authors to consider the following comments:

11)      Firstly, many words are underlined in the article. The underbars have to be removed.

22)      In the introduction at lines 104-105, the authors should precise that indoor temperature is most of the time higher than outdoor temperature IN THAILAND.

33)      In section 2.Methods, at line 124, I’m wondering if the citation number [18] is correct.

44)      In Tables and Figures in the article, the physical units have to be given like in Table 1,5 and Figure 2,3,4,… . In Table 1, is it the average wind speed at 10m ? Please give the height associated to the wind measurement.

55)      In Table 2 and at line 464 in section 3.4.2, it is mentioned that the shutters are lowered when the sun radiation is higher than 3000W/m2. This value seems to be very high, is it correct ? In Figure 9, I note that the radiation does not exceed 900W/m2.

66)      For a better understanding, I recommend to the authors to rewrite Section 2.2 and more precisely the part describing the simulation. Are the simulations only on the time interval from 8am to 3pm? This time interval [8am,3pm] also corresponds to occupied hours ? What are off-hours? At line 178, how was estimated Qv? Qv was empirically determined or measured? Please clarify this section and also add a reference for the software IESVE.

77)      At line 215, the running mean of the outdoor temperature Trm is introduced. The running mean is usually performed on which time interval: 1 hour, more ?

88)      Equation (3) at line 229 is not elegant. If I correctly understand, WF is a weighting factor that is piecewise constant on 1 hour : WF(t)=max(0,DeltaT(t)). “We” may be expressed as

We =int_{t=0}^{t=24h} WF(t) dt

I invite the authors to rewrite the equation in a correct way.

99)      In Section 3.1, are the results only on 1 day or more ?

110)   In Figure 6, I think there may be a mistake because when considering Criterion 1 the results are better for B2) Shutter only than B) including B1 and B2.

111)   In Section 3.4.2, I perfectly understand that the impact of radiation on DeltaT is much higher than the contribution of the outdoor temperature. Nevertheless, I think that the interpretation of the author “DeltaT throughout the day is not in accordance wih outdoor temperature” is too strong. I recommend to mitigate this comment. The outdoor temperature still has an influence.

112)   At line 480, “monde” should be removed or replaced by “mode”.

Author Response

Thank you very much for your review. We appreciate your comments and find they are immensely helpful. Please find our answer below in blue.

1)       Firstly, many words are underlined in the article. The underbars have to be removed.

Noted. The underlines are deleted.

2)   In the introduction at lines 104-105, the authors should precise that indoor temperature is most of the time higher than outdoor temperature IN THAILAND.

We have added this in the text at line 105.

3)    In section 2.Methods, at line 124, I’m wondering if the citation number [18] is correct.

The citation number 18 was not correct. We changed it to citation number 20 at line 125.

4)      In Tables and Figures in the article, the physical units have to be given like in Table 1,5 and Figure 2,3,4,… . In Table 1, is it the average wind speed at 10m? Please give the height associated to the wind measurement.
We have added the units in all tables and figures. Besides, we improved our charts as you can find in Figure 4, 5,6.

5)      In Table 2 and at line 464 in section 3.4.2, it is mentioned that the shutters are lowered when the sun radiation is higher than 3000W/m2. This value seems to be very high, is it correct ? In Figure 9, I note that the radiation does not exceed 900W/more          

This comment is extremely relevant. We found that the results from the original manuscript are calculated from the building when the shutters are lowered during occupied hours; the shutter were not lowered when solar radiation is higher 3000 W/m2. However, we have added the option of lowering the shutters when the incident solar irradiance is more than 300 W/m2 too to compare the results with overhangs and shutter lowered during occupied hours. Please find our revision in Section 2.3 line 202-204, Table 4, and Section 3.3.1. line 302-303.

6)      For a better understanding, I recommend to the authors to rewrite Section 2.2 and more precisely the part describing the simulation. Are the simulations only on the time interval from 8am to 3pm? This time interval [8am,3pm] also corresponds to occupied hours ? What are off-hours? At line 178, how was estimated Qv? Qv was empirically determined or measured? Please clarify this section and also add a reference for the software IESVE.

The Qv was empirically determined. We have improved our Section 2.2 and added the equation to determine Qv. However, we are not sure should the equation be added to the text because it is not the main message of the study.

7)      At line 215, the running mean of the 0outdoor temperature Trm is introduced. The running mean is usually performed on which time interval: 1 hour, more ?

According to CIBSE TM52, Trm or running mean temperature is monthly mean, which is calculated from daily mean temperature.

8)      Equation (3) at line 229 is not elegant. If I correctly understand, WF is a weighting factor that is piecewise constant on 1 hour : WF(t)=max(0,DeltaT(t)). “We” may be expressed as

We =int_{t=0}^{t=24h} WF(t) dt

I invite the authors to rewrite the equation in a correct way.

We use the equation according to CIBSE TM 52, reference [20] on page 14. If the equation is not clear, we have added another equation to explain the WF.

9)      In Section 3.1, are the results only on 1 day or more ?

It is an assessment during non-heating period.

10)   In Figure 6, I think there may be a mistake because when considering Criterion 1 the results are better for B2) Shutter only than B) including B1 and B2.

We have rechecked the simulation several times. This may have occurred from: 1) some diffused solar radiation from the overhangs,

2) The overhangs may block the wind, which affects wind speed,
3) errors from simulation

11)   In Section 3.4.2, I perfectly understand that the impact of radiation on DeltaT is much higher than the contribution of the outdoor temperature. Nevertheless, I think that the interpretation of the author “DeltaT throughout the day is not in accordance wih outdoor temperature” is too strong. I recommend to mitigate this comment. The outdoor temperature still has an influence.

Noted, we have improved our discussion here.

12)   At line 480, “monde” should be removed or replaced by “mode”

Noted. We have revised the text.

Besides the revision mentioned above, we added Figure 12 to our revision to show the impact of shading devices during the day. Discussion has been revised to appoint specific issue of a child center. The English language editing has been revised by MDPI author service. 

Reviewer 2 Report

1. The title of the paper should be revised.

2. The English should be improved, minor error in Figure 6. Results of overheating assessment using solar protection strategy “including” overhang and shutter.

3. Short paragraphs require rearrangement.

4. The units in Y-axis of Figure 5 need to be marked, check others.

5. The specific simulation time period needs to be specified, like how is the vacation?

6. Further discussion is needed on how Child Development Canters differ from other buildings.

Author Response

We appreciate your valuable comments. Please find below our answers below in blue.

  1. The title of the paper should be revised.

We have revised the title. Could you please advise more which direction to revise the name?

  1. The English should be improved,minor error in Figure 6. Results of overheating assessment using solar protection strategy “including” overhang and shutter.

Noted. We have our manuscript checked by the editing service from MDPI.

  1. Short paragraphs require rearrangement.
    Could you please elaborate on that? Which paragraphs require rearrangement?
  2. The units in Y-axis of Figure 5 need to be marked, check others.
    Noted. It is done. We have checked and added the units of Figures 4, 5, and 6.
  3. The specific simulation time period needs to be specified,like how is the vacation?
    There is no school breaks for CDCs. The simulation has been done for all seasons.

6. Further discussion is needed on how Child Development Canters differ from other buildings.

We have added on time analysis according to the schedule for the children in CDCs. This can explain the more specific experience in CDC which is different from the other buildings.

Furthermore, from the original manuscript, we found that the results of integrating shutters were from the scenario where the shutters were lowered during occupied hours (it was indicated "higher than 3000 W/m2 in the original manuscript). So, we added more scenario which is the shutters are lowered when incident solar irradiance is higher than 300 W/m2. This revision is in Section 2.3 and 3.3.1. Besides, we added Figure 12 to our revision to show the impact of shading devices during 24-hour intervals.

Round 2

Reviewer 2 Report

I have no more comment.